# Novel Approaches to Air Pollution Exposure and Clinical Outcomes Assessment in Environmental Health Studies

**Shaked Yarza** [1,2], **Lior Hassan** [1,2], **Alexandra Shtein** [3], **Dan Lesser** [3], **Lena Novack** [1,2], **Itzhak Katra** [3], **Itai Kloog** [3] **and Victor Novack** [1,2,*]

1   Negev Environmental Health Research Institute, Clinical Research Center, Soroka University Medical Center, Beer-Sheva 8457108, Israel; shaked.yarza@gmail.com (S.Y.); lior0351@gmail.com (L.H.); novack@bgu.ac.il (L.N.)
2   Faculty of Health Sciences, Ben-Gurion University of the Negev, Beer-Sheva 8457108, Israel
3   Department of Geography and Environmental Development, Ben-Gurion University of the Negev, Beer-Sheva 8457108, Israel; shtien@post.bgu.ac.il (A.S.); danless@post.bgu.ac.il (D.L.); katra@bgu.ac.il (I.K.); ikloog@gmail.com (I.K.)
*   Correspondence: victorno@clalit.org.il;

**Abstract:** An accurate assessment of pollutants' exposure and precise evaluation of the clinical outcomes pose two major challenges to the contemporary environmental health research. The common methods for exposure assessment are based on residential addresses and are prone to many biases. Pollution levels are defined based on monitoring stations that are sparsely distributed and frequently distanced far from residential addresses. In addition, the degree of an association between outdoor and indoor air pollution levels is not fully elucidated, making the exposure assessment all the more inaccurate. Clinical outcomes' assessment, on the other hand, mostly relies on the access to medical records from hospital admissions and outpatients' visits in clinics. This method differentiates by health care seeking behavior and is therefore, problematic in evaluation of an onset, duration, and severity of an outcome. In the current paper, we review a number of novel solutions aimed to mitigate the aforementioned biases. First, a hybrid satellite-based modeling approach provides daily continuous spatiotemporal estimations with improved spatial resolution of $1 \times 1$ km$^2$ and $200 \times 200$ m$^2$ grid, and thus allows a more accurate exposure assessment. Utilizing low-cost air pollution sensors allowing a direct measurement of indoor air pollution levels can further validate these models. Furthermore, the real temporal-spatial activity can be assessed by GPS tracking devices within the individuals' smartphones. A widespread use of smart devices can help with obtaining objective measurements of some of the clinical outcomes such as vital signs and glucose levels. Finally, human biomonitoring can be efficiently done at a population level, providing accurate estimates of in-vivo absorbed pollutants and allowing for the evaluation of body responses, by biomarkers examination. We suggest that the adoption of these novel methods will change the research paradigm heavily relying on ecological methodology and support development of the new clinical practices preventing adverse environmental effects on human health.

**Keywords:** exposure assessment; clinical outcome assessment; air pollution; hybrid satellite-based models; real temporal–spatial activity; low-volume air pollution sensors; human biomonitoring

## 1. Introduction

There is number of ways in which an air pollution may affect the biosphere, and human health in particular. Pollution is associated with the damaging of food and water supplies while reducing crops

yields, damaging wildlife habitats by accelerating arctic warming, triggering climate changes, increasing energy consumption, elevating morbidity burden and millions of premature deaths. According to a global burden of disease report (GBD, 2016), air pollution was ranked as the fifth leading risk factor to disability-adjusted life-years (DALY) and responsible for 7.5% of mortality cases in the world [1].

Research programs investigating the association between air pollution and human health normally encounter two major challenges of exposure assessment and health outcomes evaluation. Methods currently in use for exposure assess have several inherent problems, affecting the way clinical studies evaluate environmental impacts. The following are the common barriers for accurate estimates of the exposure burden (Figure 1): (1) Low spatiotemporal resolution relying on sparse monitoring stations might lead to misclassification bias for patients residing distantly from the monitoring sites [2,3]; (2) lack of the assessment of the indoor air quality [4,5], as most of the models dismiss the difference between the indoor/outdoor pollution and essentially use the outdoor air pollution as a proxy for overall exposure [6,7]; (3) static spatio-temporal approach to the exposure assessment based mostly on residential address and without taking into consideration individuals' mobility throughout the day [8–11].

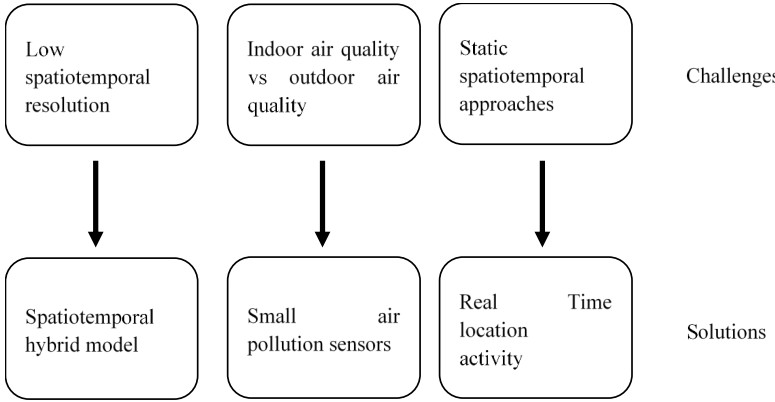

**Figure 1.** Exposure assessment challenges and their novel solutions.

Health outcomes are frequently based on medical records which poses a number of challenges (Figure 2). (1) The use of electronic medical records that might provide inaccurate data and frequently lack information on the onset and the duration of events [12–14]; (2) Seeking medical attention is frequently influenced by patients' self-perception regardless of their clinical state [15–17]. Subclinical events, i.e., events without clinical symptoms presentation, are not routinely recorded while can be clinically important [18–26]; (3) The need to obtain human tissue samples for measuring of biological indicators that can reveal a plausible biological mechanism is limited due to a small sample size prospective studies plagued by high costs [27–31].

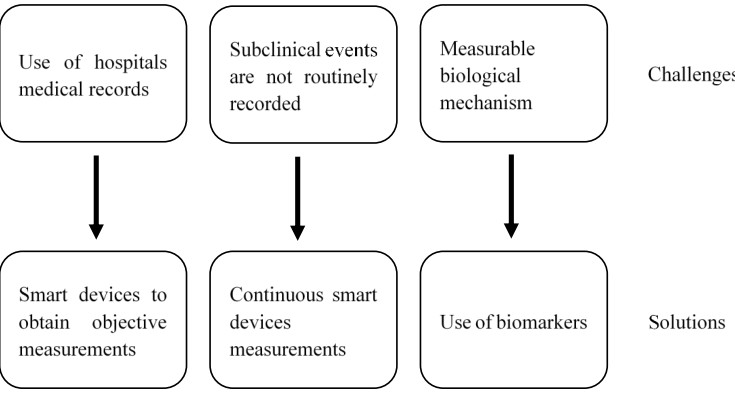

**Figure 2.** Outcome assessment challenges and their novel solutions.

We aimed to review the challenges of both exposure and clinical outcomes assessment, and present novel approaches to face these challenges.

## 2. Exposure Assessment

Accurate exposure assessment is essential for studying the association between health outcomes and air pollutants. Relying on air pollution measurements from ground-based monitors for health exposure assessment introduces exposure misclassification due to the limited spread of monitors over space and time. The monitors are usually located in urban areas where the spatial and temporal variability in air pollution is high due to variations in human activity [32,33]. These limitations led to the development of various modeling approaches which aim to provide spatiotemporally resolved estimations of different air pollutants and allow improved exposure assessment.

Air pollution exposure models can be divided to the following main classes [6,7,34], each characterized by certain merits and limitations:

(1) Proximity-based assessments that are based on the distance between patients' locations or their residential address and a certain emission source (e.g., distance to roads [35]), or on the mean of the monitors within a city [3], or simply from the closest monitor [36,37]. The strength of this approach is mainly its simplicity that does not require special expertise; however, its shortcoming is the difficulty to accurately assess the inter-urban variations;

(2) Statistical interpolation that provides estimations using geostatistical techniques (e.g., kriging [38], inverse distance weighting [39]) of air pollutants based on the measurements taken in a network of monitors. This approach might be too simplistic for urban areas that are characterized by high spatial variability of air pollutants derived by local sources. This limitation might be resolved by relying on more advanced geostatistical methods such as universal kriging that combines measured pollutants observations with modelled data [40];

(3) Land use regression models that provide air pollution estimations by calibrating the model using measurements from sparse air quality monitors. Multiple predictors (e.g., land use data, altitude, meteorology, traffic, and roads data) are used in a stochastic model that is applied to unsampled locations [41] in order to provide spatially continuous estimations. Although these models provide spatially resolved predictions, the land-use terms do not vary temporally, thus estimations from such models are relevant mainly for studying association with long term health outcomes [42];

(4) Chemical transport models (CTMs) that simulate dynamics of atmospheric pollutants by combining meteorological and chemical modules. Calculations from such models have shown good agreement with observations in some regions [43,44], however their development is usually computationally intensive, demands detailed input data, and requires expertise in meteorology and climatology to run such models [6]; and

(5) Hybrid models which provide spatiotemporally continuous air pollutants estimates using elements of land use regression that account for the local effect of spatial predictors [34] (e.g., land use data, proximity to roads and emission sources, roads and population density) alongside the effect of various temporal variables such as: meteorology, green space and measures of pollutants from satellite-based remote sensing, and pollutants estimates from CTMs. These models implement different statistical approaches to account for the relationship between pollutants measurements from ground-based devices (monitors or sensors) and various predictors. Both parametric and nonparametric statistical approaches have been used to model this relationship, allowing extending the estimates of air pollutants beyond the limited spread of measurements over space and time.

The models above relate to outdoor pollution exposure assessment methods, however, most of the population spends the majority of their times indoors [45]. Outdoor levels contribution to indoor levels depends on pollutants particles size, ventilation and buildings structures [46], yet indoor activities dominant the indoor pollutants levels [47,48]. This makes understanding the influence of indoor air pollution sources on human health crucial, but quantifying the influence of each source is challenging since information on indoor pollution levels is constructed of various possible sources such as ambient

particles penetration to homes and generated indoors particles due to raising pets, air conditioning or heating systems, smoking, cooking, candles, or wood burning and other combustion processes [49,50]. Despite numerous attempts to establish an association function between indoor and outdoor pollution, the levels of the indoor pollution (although arguably more important to the human health) are difficult to ascertain. Our group has attempted to develop such function [51], but the overall accuracy of the approach remains questionable.

## 2.1. Remote Sensing

The integration of satellite-based remote sensing data in air pollution hybrid models became widely used over the last decade, specifically for particulate matter ($PM_{10}$ and $PM_{2.5}$) [7,34], and Nitrogen Dioxide ($NO_2$) [7] assessment. Aerosol optical depth (AOD) is an example of satellite-based products available from the moderate resolution imaging spectroradiometer (MODIS) onboard the Terra and Aqua satellites. It is a measure of the extinction of the solar beam by aerosol particles in the vertical column of atmosphere over a given location and is therefore useful to estimate PM concentrations. Remote sensing data has been also used to provide a product of the total $NO_2$ in the tropospheric column measured by the ozone-monitoring instrument (OMI) on board of the Aura satellite. The strength of remote sensing products is that they constitute real physical measurements of specific air pollutants within the atmospheric column. The availability of satellite-based products from multiple satellite platforms for a sequence of several years made them useful as inputs in air pollution models that aimed to provide spatiotemporally resolved estimations allowing for exploring associations with both short and long-term health effects. Yet, the known challenge is to integrate these products in models that aim to estimate air pollution at breathing level since they provide measurements for the entire atmospheric column between ground level and the satellite. Various statistical approaches were used to incorporate satellite-based products in ground level air pollution hybrid models. One example is the hybrid mixed effects approach that was used to model the relations between ground level measured air pollutants, satellite-based products, and various spatiotemporal predictors. This approach assumes that the relations between ground level air pollution measurements and the satellite-based observations might change on a daily level, and in some cases also by region, therefore the model allows the regression intercepts and slopes of the satellite observations to vary daily and spatially. Using spatially and temporally resolved satellite-based products as a predictor allows extending the air pollution estimates to locations and time points where ground level measurements are absent. This modelling approach was used in different regions, including Europe [52,53], United States (US) [54–57], China [58], Mexico city [59], and Israel [60,61] to estimate daily $PM_{10}$ and $PM_{2.5}$ concentrations in spatial resolution of $1 \times 1$ km$^2$ and showed good performance with cross validated (cv) total R$^2$ ranging between 0.70–0.92, depending on the area and the specific methodology that was applied. Similar approach was used to estimate daily $NO_2$ concentrations in the US [62], Hong Kong [63], and Switzerland [64] with cv R$^2$ of 0.79, 0.84, and 0.58, respectively.

Ensemble modelling is another common statistical approach [64–67], that incorporates predictions from multiple base learners, which allows combining their predictive power and creating a final model that outperforms each base learner. A recent model developed for the contiguous US used generalized additive model that accounted for geographic difference to combine $PM_{2.5}$ estimates from neural network, random forest, and gradient boosting. This model showed good performance with cv R$^2$ of 0.86 for daily $PM_{2.5}$ predictions [65]. Similar ensemble modelling approach was used over Italy to estimate $PM_{10}$ and $PM_{2.5}$ concentrations using estimations from various models (mixed effects model, random forests, extreme gradient boosting and chemical transport model) showing improved model performance in comparison to the individual input models [68]. Random forest (RF) were successfully used to estimate $PM_{10}$ and $PM_{2.5}$ concentrations over Italy [67], showing promising results that outperformed previous model that was based on the mixed effects modelling approach [52].

These recent satellite-based hybrid models carried out a preliminary step of filling in using machine learning methods [52,65,67,68]. These models used simulated AOD\$NO_2$ data from chemical transport

models (e.g., GEOS-Chem, Community Multiscale Air Quality (CMAQ)) or from the Copernicus Atmosphere Monitoring Service (CAMS) Reanalysis alongside additional predictors such as land-use types, and meteorological variables, day of the year and geographical coordinates.

Some limitations and future research directions should be considered for satellite-based hybrid models. The above mentioned studies estimate daily mean (24 hour) air pollutants level using measurements taken in a spcefic time of day by sattelites with sun-syncronuch orbit. Integration of sattelite-based products from geostationary satellites that provide frequent daytime measurements might be benefical as frequent measurements can be combined and constitute a better predictor of daily (24 hour) mean air pollutants. Future research might also benefit from including satellite-based products with improved spatial and temporal resolution from the new generation of satellites (e.g., the MAIA instrument of NASA and the Sentinel-5 mission of ESA) and possibly provide additional information about the chemical composition of particles. The challenge of modelling air pollution at the ground level might be further addressed by studying the relations between ground-based measurement and products from satellites that provide information about the vertical profile of the atmospheric column. Another direction that should be taken into account in future studies is the integration of measurements from low cost air monitors that will increase their spread to rural areas or provide better spread of measurements whithin urban areas.

## 2.2. Low Cost Air Pollution Sensors

Several common methods for measurements of particulate matter are used for quantitative and modeling research. The gravimetric principle describes a quantitative determination of an analysis based on the mass of a solid (1405 TEOM Continuous Ambient Air Monitor). Such certified high-precision devices are typically large, stationary and expensive and therefore very sparsely deployed. Typically, only a few stations will cover large urban areas [69]. More fine-grained measurements are important, since exposure levels have been observed to vary even in close proximity during dust storm events. Despite the relatively nearby measurements of dust-PM by portable industrial instruments (such as the portable TSI DustTrak Aerosol Monitor) [70], the accuracy of the assessment for dust atmospheric distribution remains questionable.

The high cost of industrial, certified gravimetric instruments makes their use non-ideal for researches major, extensive research program. Previous studies considered the possibility of using commercial off-the-shelf (COTS) sensors for a more plausible economic solution, giving high temporal resolution along with a wider spatial one [71–75]. Low-cost sensors (typically mini-lasers, electrochemical sensors and metal oxide ($MO_x$) sensors, but also passive samplers) are becoming common in air pollution assessments, although some of the sensors available on the market are delivering highly questionable results. Many of these have detection levels at PPM concentrations far beyond what is found even at pollutant hot spots, which is relevant for many high-risk environments.

Khadem et al, suggested a measurement method, utilizing a node system of spatially distributed smart COTS sensors [76]. Along with a connection to a wireless network at sink nodes, such a system has the potential to increase spatial resolution significantly with limited effects on temporal resolution.

Urban measurements have also shown great promise, when the Indian government sanctioned various experiments to monitor and reduce air pollution levels in major cities (Delhi and Mumbai), including one conducted with COTS sensors [77]. The testing, which was conducted for over a week at the end of December, showed the high temporal resolution that could be gained, while additional factors were able to be taken into consideration: from junctions with slower traffic movement which caused a spike in PM counts, up to 45,000 particles per 0.01 cubic feet (PPCF) compared to the average 20,000 PPCF count, to an empirical decrease a simple face mask could provide in inhalation of said dust (from an average of 9000 PPCF in polluted areas, a decrease of over 50% to less than 4000 PPCF). The comprehensive report has also shown the association between climatological factors to changing air pollution levels, where pollution levels increased by over 400% with rising humidity levels, while temperature levels show an inverse correlation to dust levels.

A similar attempt was made when a participatory urban sensing framework for $PM_{2.5}$ monitoring with more than 2500 devices (predominantly Plantower G3) was deployed in Taiwan and 29 other countries [78]. Based on two case studies, rising dust levels were seen based on local predictions (holidays, traffic etc.) and were confirmed through mechanisms that assess the data quality based on historical data and nearby devices correlation. GIS (Geographic Information Systems) software was incorporated to spatially present all relevant data for participating cities.

A prototype called HOPES (Hone Pollution Embedded System) was created based on new indices for the indoor air quality (IAQ) criterions [79]. Under the slogan "A cheap and third-age-friendly home device for monitoring indoor air quality" the new approach based on PM and specific gases sensors is beginning to emerge. Furthermore, a recent study took personal measurements a step further, by creating a WSN (wide sensor network) that is capable of sensing different environmental factors: $PM_{2.5}$, temperature, relative humidity, air pressure and UV radiation [80]. These readings can play a significant role in personal health monitoring and protection. Through the web user interface, each user can assess her ambient environment. Personalized alarms can also be set to notify users to take timely protections when $PM_{2.5}$ value is above the recommended value.

Exposure assessment based on the mobile phone devices is an another promising direction Outdoor mobile measurements with COTS are still in the works, but more professional instruments have already been used on different platforms (i.e., car, van, pedestrian, bicycle, tram, airship) for mobile monitoring of the various properties of PM. In particular, mobile air quality monitoring is mainly carried out through motor vehicles that can be equipped with voluminous and moderately heavy instruments [81–85]. This poses another issue in data reliability due to pollution caused by said vehicles. To alleviate the disturbances caused by motorized vehicles, Hankey et al. used a bicycle-based system to develop different LUR models for particulate concentrations [86]. A similar attempt was made by Thai et al, who used an instrumented bicycle to elucidate particulate matter exposures over a 14 day period [87].

### 2.3. Real Time Location Activity and the Use of Smart Devices

Human activities throughout the day determine the burden of individual air pollution exposure. The great variability in human activities, together with their time spent indoors, at work or in traffic, makes the accurate determination of their exposure to air pollution extremely difficult [8,9,88,89]. New methods for ambient air pollution exposure assessment in epidemiological studies were developed in recent years, but they are still predominantly based on the static temporo-spatial approach. Thus, the most prevalent method to estimate personal exposure is based on residential addresses [7]. This method is prone to bias due to a person's mobility throughout the day that may result in variability in exposure and thus lead to an inaccurate estimation when only home address-based concentrations are taken into account [10,11].

A study conducted in the United States (US) examined the association between residential mobility during pregnancy and possible exposure misclassifications in birth defect studies [90]. The study showed that maternal residential mobility patterns during pregnancy were associated with various socio-demographic characteristics. Their results suggest that birth defects studies that use maternal residential addresses may be subjected to non-differential exposure misclassification. Another study carried out in the US examined the effect of residential mobility of children on estimates of traffic-related air pollution (TRAP), satellite estimated greenspace and socioeconomic characteristics [91]. Estimations based on birth and last known addresses compared to annual address history revealed a significant exposure misclassification. Similarly, disregarding the type of individual daily activity may result in exposure misclassifications. Staying in indoor workplaces, industrial or recreation centers, and driving or spending time in traffic while exposed to TRAP causes the variability in exposure to air pollution [92].

These limitations demonstrate that valid and updated information on the temporal-spatial distribution correlated with the timing and the levels of exposure to air-pollution by real time location,

is crucial for the estimation of environmental pollution exposure. So far, the proposed solutions for achieving this level of accuracy were either too expensive or labor intensive.

One of the modern necessities of almost every human today is an ownership of a smart phone. Almost every citizen in developed countries carries a smartphone to help in navigating, communicating, acquiring knowledge and improving the quality of life [93]. According to Ericsson Mobility Report, in 2016 there were 3.8 billion global smartphone users with estimated growing numbers of up to 6.8 billion users by 2022 [94]. The growing number of smartphones can lead to the development of computerized methods to assess individuals' exposure to air pollution with more precise temporal-spatial resolution [95]. GPS information can also determine the velocity of individuals which can detect whether an individual is in traffic, located at work or sleeps at home, and by that, differentiate their exposure while in different activities. Real-time location tracking using smartphones enables analysis of human behavior and activities, minimizes uncertainties due to human mobility during exposure assessment.

Studies carried out in both Belgium and the US examined the exposure to $NO_2$ [96] and $PM_{2.5}$ [97] pollutants concentrations using a dynamic approach based on mobile devices and cellular networks to solve the spatial distribution and compared the results to the traditional static residential addresses evaluation method revealed significant differences between two approaches in the exposure quantitation. Another study conducted in the US on a large population scale, demonstrated that measurements of exposure to $PM_{2.5}$ pollution based on data location derived from mobile devices, compared to data based on residence-only, results in more precise estimates of the particular matter [98].

Time activity patterns assessed by GPS tracking also serves as a tool for advanced personal spatiotemporal exposure assessment models [99]. GPS based individual exposure assessment models evolved from extracting spatiotemporal raw data from daily land-use individuals' mobile devices to analyzing GPS geo-spatial data and data mining trajectories by highly sophisticated algorithms for outdoor and indoor microenvironments [100]. Assessment of time activity patterns using GPS analysis was also utilized in an experimental prospective study in which exposure to ultrafine particles (UFP) was evaluated in 24 couples (same residential address) of full-time working men and homemaker women, suited with particle counter and GPS monitor [101]. This study found different levels of exposure thus demonstrating the feasibility of the personal GPS monitoring.

Assessing time activity patterns can also be done using mobile devices such as Garmin GPS receivers that can record participants' location several times a day. These devices were used to evaluate the association between the levels of exposure and the roads type or mode of transportation used [102]. Moreover, integrating the data from the devices with pollution sensors and objective clinical measurements gives insight about spatio-temporally resolved levels of pollution and the health effects of the exposure [103,104].

Mobile phones in recent years are increasingly used as a technological basis for air pollution sensing devices [105]. Using low-cost mobile stations connected to mobile phones enable acquisition of large data quantities and can be used by a large population in a relatively cost-effective manner [106,107]. This method of exposure assessment can provide data on the personal exposure to air pollution and help get high spatio-temporal resolution on urban air pollution assisting for example in municipal planning [108,109]. The limitation of this method is that mobile sensing devices provide data with missing values in both time and space. The missing values can be due to loss of battery power [110], loss of signal in indoor environment [100] or device misuse [111]. Dealing with missing GPS data is challenging and can be handled by methods such as data filtration like selecting days with fewer missing data, smoothing techniques [112], interpolation and imputation of the data [113]. Mendes et al., approached this limitation using kriging technique for data interpolation, showing superiority of this method to the traditional ones [114].

### 3. Assessment of Clinical Outcomes

Evaluating the outcomes of exposure to air pollution is frequently based on administrative data from health institutions, yet the variability of the medical coding, and the availability and accuracy of the health data, have well known limitations.

The most common method to evaluate the incidence of outcome in environmental studies is based on the events in which the subjects have sought medical attention (e.g., hospital visits or outpatients' clinics visits). However, assessing the clinical outcomes using medical records poses several problems. First, medical attention is frequently delayed by the patients and is influenced by the patient characteristics such as low medical literacy, social network characteristics, socioeconomic status, severity of the condition and availability of medical services [15–17]. In addition, medical records might lack information on the evaluation of the onset and the duration of the event, e.g., in of the analysis of atrial fibrillation (AF) it is important to determine the start and the duration of the event [12–14]. Finally, subclinical events, i.e., events without clinical presentation such as elevated glucose [25,26], elevated blood pressure [23,24], subclinical AF [20,21] etc. [18,19,22], are not routinely recorded. These limitations can narrow the outcome assessment and introduce bias in the estimation of the effects of air pollution on health outcomes.

### 3.1. Utilization of Smart Devices

In recent years there has been a growing use of smart devices and wearable devices for assessing clinical health outcomes in different fields. The devices can be used to detect the development of diabetes [115], to monitor glucose levels [116] and levels of glycated hemoglobin [117], to record symptoms related to Chronic Obstructive Pulmonary Disease (COPD) [118] and to improve blood pressure control using self-management systems [119]. These studies have demonstrated how patients' health can be improved by incorporating smart devices into medical practice, yet frequently the use of these devices relays on the patient collaboration and greatly depend on unbiased patient reports that can result in misclassifications of the outcome [117].

In the field of cardiovascular medicine, the implantable cardioverter defibrillator (ICD) and pacemakers can be used for detecting cardiac arrhythmias since these devices are capable of recording and analyzing abnormal heart rate events. Several studies have examined the importance of these devices for the detection of new onset of AF [120] and asymptomatic AF [121,122]. However, the inherent limitation of the health information provided by these devices is that it is pertinent to a selected group of patients in whom the use of these devices was clinically indicated.

Smart devices such as wrist bands and smart watches have been used for assessing abnormalities in cardiac rhythm [123], AF and asymptomatic AF [124], without the constrains of the subjective patients reports and without the need to constantly validate the data. A recent study used smart Apple Watches to detect cases of an irregular heart pulse in the general population, thus creating for the first time a prospective large-scale study that identifies subclinical events [125].

However, the use of the smart devices has a number of limitations: first, the efficacy of the smart watches for detecting clinical events was still not evaluated in clinical trials; furthermore, patients are frequently non-compliant with the requirement of fulfilling personal diaries, unlike in traditional studies during which patients are closely monitored for all clinical events. Additionally, use of smart devices requires a certain degree of technological literacy, thus limiting their use to younger population groups. Another limitation is the smart phones' energy consumption and the need for frequent charging [126,127], which needs to be addressed for example by the development of the energy-efficient algorithms [128] In addition clinical and physiological data ascertainment via mobile phones poses a number of challenges: (1) Accurate, "medical grade" data acquisition. Only lately FDA approved devices capable of providing ECG tracking; (2) Data processing and optimization, which requires threshold-based methods [129]; (3) Data transfer which is usually done by third-party systems which might result in random and partial data transfers. This can be addressed by applications that provide fixed transfers times [130]; (4) Data storage, especially of the large volume longitudinal signals

such as blood pressure, ECG etc. An efficient solution requires cloud storage services [131]. Lastly, although not within the scope of the current manuscript, the serious issues of the health data privacy must be addressed on the legislative level. Missing values of measured clinical parameters such as heart rates and ECG can be a consequence of lack of charging and misfunctions of the devices due to technical problems [132]. This can be handled using methods such as data interpolation [133] and imputation [134].

*3.2. Human Biomonitoring*

The underlying assumption of ecological research is that ambient outdoor air pollution is a valid proxy of the true individual levels of exposure. Nevertheless, the health effect of an ambient exposure is frequently confounded by socio-economic status, occupation, smoking and other factors, that are hard to account for in a standard analysis, resulting in spurious or biased associations.

We believe that measurable biological indicators sampled from human tissues are required to validate the current ecological methods of exposure assessments. Furthermore, this approach will extend our insight on the plausible biological mechanisms which contribute to the pathophysiology of the environmental health effects.

Biologic indicators or biomarkers include biochemical, molecular, genetic, immunologic, or physiologic signals [135]. Human biomonitoring (HBM) is frequently used worldwide to estimate the extent of exposure of populations to potentially dangerous chemical substances [136]. However, HBM studies have been used mainly in the occupational-health field [137–139]. Yet, recently the focus has changed towards the assessment of an environmental exposure in the general population. For instance, an HBM of environmental chemicals in the Canadian Health Measures Survey is a comprehensive initiative providing general population HBM data in Canada. It is an ongoing cross-sectional survey implemented in 2 year cycles with an enrollment of up to 7000 people in each cycle. Its recent 2016 report presented the information on biomonitoring results for 279 chemicals; approximately half of the chemicals measured in individual's blood and urine samples were detected in more than 60% of the samples [140].

The Center for Disease Control (CDC) and Prevention have been providing similar information on the nationally representative biomonitoring data in the general population in the US since 1999. The CDC reports present analyses of blood, serum, and urine samples from random subsamples of 2 year surveys (National Health and Nutrition Survey–NHANES), typically including up to 6000–7000 participants at each round, and an extensive list of chemicals tested [141].

Biomarkers have an additional role related to environmental health research; validation of the causal relationship between air pollution exposure and health outcomes by revealing of a plausible biologic mechanism. Studies have examined the effect of pollutants on biomarkers expression in different human tissues such as adipose tissue [29], cardiomyocytes [30], lung epithelial cells [31], and blood [28]. For instance, the Multi-Ethnic Study of Atherosclerosis (MESA) showed that the $PM_{2.5}$ exposure was correlated with an increase in blood concentration of C-reactive protein (CRP), fibrinogen and E-selectin [142], markers related to the atherogenic process [143–145], suggesting a plausible biological mechanism by which air pollution could contribute to the development of cardiovascular diseases.

The main problem with conducting such a survey is a potential bias introduced through the sampling design and the participant selection, e.g., subjects with higher socioeconomic status and women are more likely to participate in such studies [146]. The WHO recommends a minimum of 120 randomly selected individuals per population group to allow for the estimation of group-specific reference values with sufficient precision and meaningful comparison of population groups in surveys. Additionally, the high cost for obtaining and analyzing each sample in such studies is extremally high, ranging from less than €100 for metals to €800 for dioxins [147].

Population Biomonitoring

We recently have developed an innovative cost-effective approach to population biomonitoring based on the national blood banking system. The method is based on analyzing the blood donations collected and processed daily in Israel.

The use of blood from blood donors for research objectives is previously described; e.g., the Danish Blood Donor Study (DBDS), a large-scale research project and biobank established in 2010 [148]. Although numerous studies were conducted via the extensive database of blood donors, showing associations between risk factors, biomarkers, health effects, and diseases, to this date, the use of the blood banking for the human biomonitoring is yet to be widely established.

Based on the established blood bank system, blood samples of donors can be tested for the detection of different pollutants. Per National Blood Bank regulation, the serum separated from the unit of the blood is to be stored for the period of one year. The donors are routinely consented to the testing of the stored plasma. Levels of the pollutants' concentration levels can then be linked with the air pollution and meteorological data assessed at the location of blood collection services (short-term exposure) and at the donors' permanent address (medium and long-term exposure), by using satellite-based exposure models. The sampling method of donations used for such studies can be aimed to create geographically representative cohorts for different areas. Repeated testing for recurrent donors in certain regions will provide indication on changing trends in exposure in these areas.

This method has a number of advantages. First, the tested population comprised healthy volunteers who represent an ideal population for the precise assessment of the environmental exposure, as they are not treated with medications and conduct an active lifestyle; second, 80% of the blood donors are returning donors, thus, repeated testing for recurrent donors in the population will provide indication on changing trends in exposure in the area; lastly, as blood donation stations are spread all over the country, the population of donors will serve a geographically valid proxy for the general population. The method has some challenges as well with the main being the use of the collection and storage equipment not specifically designed for the measurement of the low concentrations of the potential pollutants, i.e., possibility of the contamination. In addition, the specific medium to be used for the pollutants testing can be different from the usual standard for blood banking. For instance, our preliminary experiment demonstrated that only whole blood would reflect the actual concentration of some of the heavy metals in the blood, while only the serum is routinely collected and stored in the National Blood Bank.

The important consequence of such studies is the establishment of a framework of dynamic national biomonitoring by repeatedly assessing the exposure to selected chemicals and pollutants. In the future, daily assessments of potentially hazardous elements in the donated blood will allow to map the areas of exposure. Furthermore, the information collected in proposed surveys can lead to a new paradigm of tailored environmental preventions measures.

## 4. Conclusions

The environmental health research faces the inherent problem of the balance between the desire to assess a population on a wider scale and ability to obtain an accurate exposures and outcomes data. So far, we have been forced to sacrifice one goal to achieve another; the researchers could assess large populations using an ecological design with imprecise exposure assessment of conduct personal exposure studies on a small scale. Latest development in the field of the exposure assessment together with the ability to access detailed clinical data is changing the research paradigm. Estimation of real-time location-activity using GPS tracking of cellular phones together with a high-resolution hybrid satellite model that can measure pollutants levels with a spatial resolution of up to $1 \times 1$ km$^2$ allows for a precise spatio-temporal solution. Wearable devices will provide continuous real time clinical measurements altering the way the health data is being collected and assessed. Smart devices combined with real time location activity retain information on exact locations and timeframes in

which populations were exposed to different hazards. Human biomonitoring samples can help better understand both the dose of pollutants the body absorbs and the plausible biological mechanisms which contribute to the health outcome. Finally, the use of advanced computational and statistical approaches incorporating the elements of the machine learning can provided the new analytical framework. We believe that the described approaches will allow us to develop novel diagnostic and treatment tools to address the environmental effects on human health, which in turn, will be adopted by the clinical community.

**Author Contributions:** Conceptualization, L.N., I.K. (Itzhak Katra), I.K. (Itai Kloog), and V.N.; investigation, S.Y., L.H., A.S., and D.L.; writing—original draft preparation, S.Y.; writing—review and editing, L.N. and V.N.; visualization, S.Y.; supervision, V.N. All authors have read and agreed to the published version of the manuscript.

**Funding:** This research received no external funding.

**Conflicts of Interest:** The authors declare no conflict of interest.

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
