# Peer review of "Novel Approaches to Air Pollution Exposure and Clinical Outcomes Assessment in Environmental Health Studies"

_atmosphere, doi:10.3390/atmos11020122_

Round 1

Reviewer 1 Report

This manuscript by Yarza et al brings to the readership of Atmosphere a thorough and well presented review of the latest developments in the field of personal environmental exposure assessment and continuous detailed monitoring of clinical parameters of relevance. The manuscript revises the developing new research framework in environmental health where remote sensing benefiting from satellite data and advanced modelling coupled with the strengths of small volume sensors and GPS data from smartphones is able to provide high resolution personal exposure. In parallel, it gives a critical appraisal of the growing use of smart wearable devices for assessing clinical health outcomes in various fields and the potential for biomonitoring through the assessment of several chemicals in patients' biological samples.

Minor Points for Consideration

1.  Elaborate more on the limitations of GPS tracking such as the loss of signal in indoor environments and how to treat missing values.  

2. Elaborate more on the limitations of smart devices to monitor clinical parameters such as the need for frequent recharging, data storage and transfer of data issues, missing data and how to treat missing values.     

Reviewer 2 Report

Abstract:

The authors state that the association between indoor and outdoor pollution is not fully elucidated. However, it is well established that indoor sources in most cases dominate indoor levels, as well as it has been shown in several studies that indoor levels are better proxies for personal exposure than outdoor levels, see e.g. Sorensen et al. (2005). Many researcher have tried to establish indoor – outdoor relationships for exposure assessment, but generally this turns out to be impossible since indoor sources dominate and they vary from one home to another following pattern that in most cases (and surely in most epidemiological studies) cannot be modelled using the available information. Only parts of the information would be obtainable using massive resources – this concerns information about the specific home for each individual regarding ventilation system, source of domestic heating including use of wood stoves, candlelight burning, environmental tobacco smoking, presence of pets and many other – just to list some of the most important. Moreover, one may argue that this is not really important in epidemiological studies, since what is usually determined in epidemiological studies are relationships between outdoor pollution levels and various adverse health effects. Since such relationships can be established, health effects are related to outdoor levels even when these contribute only marginally to indoor levels.

The authors speak about low-volume air pollution sensors – this is not a common term (at least to know to me) and it is not clear from the text what is meant by this phrase. Low-volume samplers are common in sampling of e.g. PM2.5 and PM10 particles, these are typically sampling for hours to days on filters for gravimetric determination of concentrations. Low-cost sensors (typically mini-lasers, electrochemical sensors and metal oxide (MOx) sensors, but it may also be passive samplers) are becoming common is air pollution assessments, although some of the sensors available on the market are delivering highly questionable results, especially at the pollutant levels in ambient air; many of these have detection level at ppm concentrations far beyond what is found even at pollutant pot spots.

Sorensen, M., Andersen, H.V., Loft, S., Raaschou-Nielsen, O., Skovgaard, L. T., Knudsen, L., Nielsen, I. V., and Hertel, O., 2005. Personal Exposure to PM2.5 and NO2 in Copenhagen: relationship to bedroom and outdoor concentrations covering seasonal variation. Journal of Exposure Analysis and Environmental Epidemiology, 15, 413-422

Introduction:

The authors list barriers for accurate exposure assessment. In relation to this they speak about low spatiotemporal resolution relying on meteorological stations leading to misclassification due to distance from patients home to these stations. Air pollution monitoring stations are not meteorological stations, although some of these include a meteorological mast to provide meteorological data for analyses (common for urban background and regional stations). Furthermore, it is not given that this leads to misclassification. This depends on the air pollutant in question and whether the pollutant has high spatial variation that cannot be resolved by the measurements at the monitoring site.

The authors talk about the lack of assessment of indoor source - I have already commented on this in my comments for the abstract. Here the authors refer to the study of Pope et al. where routine monitoring data are used successfully in epidemiological studies. This reference is to my knowledge not disputing the short-come of not including indoor sources, and since they find statistically significant relationship with outdoor air, this seems not to be a proper use of reference.

The authors talk about use of models in exposure assessment, but this description is far from covering what is currently applied in various studies. 1) Proximity-based assessments – like distance to trafficked road. One might argue that this is a model, but it would not usually be what people have in mind when talking about exposure assessment models. Here it would be more natural to discuss exposure proxies. The authors claim that the main short-come is this methods missing ability to characterise inter-urban variation, but several studies have demonstrated that this simplistic method starting with the Dutch studies by Hoek and co-workers works very well. 2) Statistical interpolation based on measurements from monitoring networks. The authors call this method too simplistic, which seems reasonable although this depends on the specific case and the amount of data available. In the work performed by Philipp Schneider and colleagues at NILU under the EU Citi-Sense project, intelligent interpolation and data fusion. 3) Land-use regression models – here the authors rightfully point at the short-come regarding temporal resolution. However, this methodology is generally used in long-term exposure assessments where the focus is on the spatial distribution. 4) Dispersion models that rely on Gaussian plume equation, which the authors claim to have unrealistic assumptions about dispersion. This can strongly be disputed, but it is also a rather uncommon approach, whereas integrated model systems can be very strong tools see e.g. Hertel et al. (2013). The latter might, however, be what the authors have in mind with 5) Integrated emission-meteorological models, which is a strange term for transport-chemistry models. 6) Hybrid models seems to be a mix of different combined assessments like the data fusion, and combined land-use regression models and transport-chemistry models, but the description is not precise.

Hertel, O., Jensen, S.S., Ketzel, M., Becker, T., Peel, R.G., Ørby, P.V., Skjøth, C.A., Ellermann, T., Raaschou-Nielsen, O., Sørensen, M., Bräuner, E.V., Andersen, Z.J., Loften, S., Schlünssen, V., Bønløkke, J.H., and Sigsgaard, T., 2013. Utilizing Monitoring Data and Spatial Analysis Tools for Exposure Assessment of Atmospheric Pollutants in Denmark Pp 95-122 (In Occurence, fate and impact of atmospheric pollutants, Ed. Laura McConnell, Jordi Dachs, and Cathleen J. Hapeman, 270 p). http://pubs.acs.org/doi/book/10.1021/bk-2013-1149

Remote sensing:

The authors talk about integration of remote sensing data into air pollution modelling as a recent development. However, the Dutch LOTOS-EUROS model developed by Peter Builtjes and colleagues at TNO has worked with data fusion of particle and nitrogen dioxide pollution for a couple of decades, so this is hardly a new development. The critical point is that most satellite data provide information about the total column from ground and up to the satellite, and the challenge is then to determine how this is distributed in the vertical.

It is unclear to me what is meant by mixed effects modelling approach and what the authors aim for when talking about the relationship between measured air pollution and satellite-based products!? When the satellite provide data for nitrogen dioxide – this is just as much a series of measurements as the ground based ones!

In the block talking about imputation of missing values in relation to satellite data, the authors talk about imputing observations. This is a wrong term – when generating data for missing values, one is not producing observations but data to fill holes in time series/sets of observations.

Small-volume air pollution sensors:

It is not common and logical wording to call a TEOM instrument for a sensor. It is an instrument, monitor or device, whereas a sensor is a part of an instrument that detect something.

The authors talk about commercial off the shelves sensors, but the list a series of papers regarding remote sensing data including optical depth measurements, and a paper about misclassification, which appear a bit strange.

The authors refer to a citizens science based study in Taiwan where PM2.5 monitoring was performed using more than 2,500 devices. However, the authors do not discuss the type and quality of device.

Real time location activity and the use of smart devices:

Here the authors discuss the possibilities of using smartphones in exposure assessment. Here it would be natural if the authors referred to mobile phone devices with air pollution sensors that have been available on the market for a couple of years. The mobile phones as such may provide useful information about time-activity pattern, and this would be natural to discuss in this section.

The authors do not refer to studies where data from mobile devices like Garmin devices and use in analyses of air pollution exposure.

The authors talk about the limitations of cohorts of people using smart devices are biased in relation to distribution on socio-economic status, which may clearly be an issue although less marked in Northern European studies.

Human Biomonitoring:

The authors mention ambient exposure being confounded by socio-economic status – this phrasing is strange. The confounding is in relation to determining exposure-effect relationships and not in determining the exposure!

Again the authors’ point as biases on socio-economic status – in studies in Northern Europe this is not a major issue, but it may of course be the case in Israel.

Population Biomonitoring:

Here it would be natural to refer to the Danish Blood Donor study, where 50,000 Danes regularly deliver blood and sub-samples are used for biomonitoring of various health indicators, exposure markers etc.
